# Exploring the Relationship between Transport Interventions, Mode Choice, and Travel Perception: An Empirical Study in Beijing, China

**DOI:** 10.3390/ijerph17124258

**Published:** 2020-06-15

**Authors:** Aihua Fan, Xumei Chen

**Affiliations:** 1Key Laboratory of Transport Industry of Big Data Application Technologies for Comprehensive Transport, Ministry of Transport, Beijing Jiaotong University, Beijing 100044, China; 15114229@bjtu.edu.cn; 2School of Traffic and Transportation, Xuchang University, Xuchang 461000, China

**Keywords:** information intervention, public transport service improvement, mode choice, travel perception, process model

## Abstract

Transport interventions help to facilitate the sustainable travel behavior. The effects of transport interventions on travel choices have been addressed extensively. However, little research has been devoted to the influence of transport interventions and travel choice on travel perception. This study aimed to investigate the relationship among the three aspects. Two intervention measures, information intervention and public transport service improvement, were selected. Intervention experiments were designed to collect mode choice and corresponding travel perception in different experiment stages. Process models of information intervention and public transport service improvement were proposed. The results show that information intervention only had a minor effect on mode choice and had no direct effect on travel perception. Public transport service improvement in in-vehicle time and comfort enhanced public transport use dramatically. Comfort improvement also had positive effects on travel perception. Walking had positive and public transport trips had negative effects on travel perception. For travelers who had a high evaluation of car trips, the probability of green mode use would decrease. Travelers who gave high marks to trips by green mode would have a higher probability to keep traveling by green mode. This study contributes to facilitating public transport use and enhancing positive perception during traveling.

## 1. Introduction

Transport interventions can help to change travel behavior and shape travelers’ sustainable behavior. Transport interventions include soft measures, such as the dissemination of green mode traveling, and hard measures, such as license plate restriction. The implementation of transport interventions facilitates travel choice change, including mode choice, departure time choice, activity choice, etc. Travel choice has impacts on travel perception. Transport interventions brings direct or indirect effects on travel perception through influencing travel choice, and, the other way round, feedback exists among transport interventions, travel choice, and perception. If the traveling experience become worse after the travel choice change, travelers will continue adjusting their choices to obtain higher levels of travel perception. Transport managers also need to adjust intervention strategies according to travelers’ perception and travel choice behavior change. The relationship among transport interventions, travel choice behavior, and travel perception is shown in Figure 1.

Transport interventions could facilitate sustainable travel behavior to some extent. The evaluation of transportation interventions should also focus on travelers’ perception change besides the change of travel choice behavior. At present, numerous studies have been performed to explore the influence of transport interventions on travel choice behavior. Rose and Ampt (2001) [1] evaluated Australia Travel Blending Project which was proposed to address environmental concerns through encouraging car use reduction and Adelaide City was selected as the case study. The results demonstrate that car driver kilometers, trips, and total hours spent has declined by 10%, 14%, and 20% separately. The Liverpool Hospital Travel Plan was initiated in Sydney, Australia, to encourage “Cycling to Work” and “Walking to Work” by providing facilities such as bike racks, showers, and lockers. Petrunoff et al. (2016) [2] collected travel data for four years during implementation of the travel plan. The data analysis results show that the proportion of staff driving to work reduced from 83% to 70% and more staff went to work by public transport, bike, or walking. However, a smaller number of studies has focused on the influence of transport interventions and travel choice on travel perception and addressed implications of travel perception for transport policy [3,4,5].

The objective of this study was to investigate the relationship between transport interventions, mode choice, and travel perception, which was used to provide recommended policy implications for sustainable travel behavior development. Soft measures including information intervention and public transport service improvement were selected as two intervention strategies in this study. The remainder of this study was organized as follows: Section 2 presented literature review, Section 3 provided methodology, followed by result analysis, and the final section concluded the study.

## 2. Literature Review

### 2.1. Transport Interventions

Table 1 presents the main strategies of transport interventions. Physical change, legal policies, economic policies, and information and education were four main strategies of transport interventions [6]. According to the intervention type, transport interventions can be divided into hard and soft measures. Hard measures force travelers to reduce car use through, for instance, increasing costs for car use and restricting car use. Soft measures that use information dissemination, education, newly developed transport services, such as shared mobility, or improvement of transport services to guide travelers to voluntarily switch to sustainable travel behavior [6,7,8]. Compared to soft measures, it was difficult to change car users’ habit and attitude towards sustainable travel mode through hard measures. It can even increase the resistance of travelers, unless these hard measures lead to positive outcomes [9]. Actual effective interventions should bring change in travelers’ attitude and further facilitate voluntary travel behavior change, instead of forced change against people’s voluntary principles [6]. For example, in reports by Susan et al. [10] and Adam and Susan [11], it showed that shared mobility promoted reduced household vehicle holdings, less vehicle miles traveled, and increased use of active transport modes. In this study, two kinds of soft measures, information intervention and public transport service improvement, were selected as the transport intervention strategies.

#### 2.1.1. Information Intervention

Information intervention in the transportation field was mainly implemented to promote and encourage pro-environmental travel behavior [12]. Knowledge was a prerequisite for the development of pro-environmental attitudes, which would further promote pro-environmental behavior [13]. Travelers’ perception of various transport modes and travel services may not be accurate and comprehensive. Information intervention was to influence a traveler’s attitude and cognition through information and knowledge dissemination and thus affected travel behavior. Geng et al. [13] analyzed the impact of information intervention on mode choice of urban residents with different travel goal frames (hedonic, gain, and green). Intervention information included six aspects, including the health benefits of riding a bike, the convenience and safety of walking, riding, and taking public transport, listing the individual cost of car use, listing the social cost of car use, addressing environment pollution caused by car use, and showed green travel campaigns and activities. The results show that travel times by green modes increased significantly for three clusters, but no significant decrease in car use. Guo and Peeta [14] studied the impact of personalized accessibility information on residential location choice and travel behavior after relocation. The treatment group was provided neighborhood accessibility information about the ease of access to destinations of different purposes before relocation. The results demonstrate that personalized information help residents find relocations that had higher accessibility to their destinations, and residents who planned to relocate would be more likely to reduce car use. Ahmed et al. [15] used air quality-based information to make school travel healthier and more environmentally friendly. A computational model was developed to help detect walking/cycling school routes that made students exposed to less air pollutants. The results show that the intervention information facilitated more students to choose active travel mode and the suggested routes. Brakewood et al. [16,17], Brazil et al. [18], Litescu et al. [19], Xiong et al. [20], and Thaithatkul et al. [21] also proved that information related to traffic and fuel cost had impacts on travel behavior, such as mode share, new car sales, energy saving, and carpooling system adoption.

In this study, information intervention experiment was designed to disseminate information about “health and environment benefits brought by non-motor vehicle traveling”, “air pollution caused by motor vehicle traveling”, and “improvements of green mode service”. Based on the information intervention experiment, the relationship model among information intervention, mode choice, and travel perception would be proposed.

#### 2.1.2. Improvement of Public Transport Service Level

Public transport plays an important role in contributing to urban sustainability. Gustavo Petro, the mayor of Bogotá, ever said “A developed country is not a place where the poor have cars. It’s where the rich use public transportation”. Attractive public transport service was the contributing factor of a sustainable transport system [22]. In recent decades, various measures of improving public transport service have been implemented to increase public transport mode share. Frequency and personal security on the bus were proved important to increase patronage [23]. Travel time, reliability, and comfort in vehicle also had a direct effect on public transport use [24]. Nurdden et al. [25] confirmed that reduced travel time, access time from home to public transport station, and fares were key factors to encourage public transport use. Accessibility had a strong impact on public transport mode share, but differences existed among different income groups and regions of varying size [26].

In this study, considering the whole trip, accessibility, next bus service, bus lane, optimization of public transport network and operation would be included to reduce access–egress time, waiting time, in-vehicle time, the number of transfers, and increase comfort in vehicle.

### 2.2. Travel Perception Measurement

Travel perception was used to describe travelers’ overall cognitive evaluation and emotional feeling during traveling. Its synonyms included travel satisfaction [27], travel feelings [28], travel liking [29], travel happiness [30,31,32], etc. The perspectives and measurement methods of these travel-related evaluations were different. Travel perception placed extra emphasis on traveler’s trip evaluation from the perspective of the traveler’s sensory feeling.

Compared with 20 years ago, with the increasing popularity of the automobile, the acceleration of urbanization, and improvement of material and spiritual pursuits, people’s daily activities became more diverse. Traveling for different purposes, such as work, school, shopping, socializing, and entertainment, has become an integral part of people’s daily life. With more understanding of activity-based travel, scholars thought that travel itself should not only be treated as ways to conducting activities, but as a significant activity [33,34]. Besides influences on activity, experience during traveling also had significant impact on people’s daily happiness and long-term life satisfaction. The impact of travel on happiness has been fully demonstrated by Ettema et al. [35]. Delbosc and Currie [36] proved that the relationship between the number of activities cannot do because of transport problem and well-being was negative.

So far, research on travel perception considering affective emotion was mainly concentrated in developed countries such as the United States, Canada, and Europe [30]. Little attention has been paid to travel perception in developing countries such as China. According to existing research, the travel perception measurement method was developed to measure cognitive judgement and affective evaluation separately or measure these two aspects simultaneously. *Affective Balance Scale* [37], *Swedish Core Affect Scale* [38], *Positive Affect and Negative Affect Scale* [39], and *Net Affect Score* [31] were methods to measure emotional well-being during traveling. Asking respondents questions about specific assessment contents was the general method to measure cognitive evaluation. Satisfaction with Travel Scale (STS) to measure travel-specific domain well-being was proposed by Ettema et al. [40]. The STS included three-item cognitive evaluations and six-item affective evaluations, as shown in Table 2.

Because of its comprehensive and clear evaluation for traveling, STS has been widely applied to evaluate travel experience. In applications, scholars usually made some changes for STS to make it better understood or reduce the burden of participants. In this study, STS was used to measure travel perception. To reduce participants’ burden and the similarities of some items after translating into Chinese, three items including “Travel was low-high standard”, “Bored—Enthusiastic”, and “Time pressed—Relaxed” were dropped.

### 2.3. Methods for Analyzing the Impacts of Transport Interventions on Traveling

There were numerous studies analyzing the influences of transport interventions on mode choice. Indicators such as change of car use kilometers, car trips, total hours spent [1] and the proportion change of car use [2] were used to evaluate the impacts of transport interventions on mode change in the long term. In the short to medium term, Braun et al. [41] applied conditional logistic regression to examine the effects of travel demand incentives on bicycle use. Friman et al. [42] studied how a temporary free public transport intervention affected car use. Structural equation model (SEM) was used with latent variables such as attitudes included. Li and Lu [43] proposed mixed logit models to estimate the effects of congestion pricing on mode choice behavior considering automobile use habit heterogeneity. Very limited research focused on the influence of transport interventions and travel choice on travel perception [3,4,5].

There were multi-stage interactions among transport interventions, mode choice, and travel perception. The implementation of transport interventions had impacts on mode choice, and mode choice and transport interventions affected travel perception. Travel perception gave feedback to mode choice in the next trips and intervention policies. Transport intervention policies responded to mode choice results and travel perceptions. Thus, the process model was introduced to analyze their interaction process among transport interventions, mode choice, and travel perception. The process model has been applied in multiple fields. It is used to track or describe the detailed process of an event. A process model is roughly an anticipation of what the process will look like [44]. The goals of a process model are to be descriptive, prescriptive, and explanatory. Firstly, through tracking the development of an event, related managers try to understand the starting point, intermediate process, and result of the event in detail. Furthermore, from the perspective of external observation, the development process of the event can be analyzed well and improvements can be made to enhance the event’s execution efficiency. Finally, the desired event process can be determined, and the results of the event can be reasonably explained.

“Process” can be a real-time process or an interactive process of different factors/events. For example, occupation, income, and family structure have impacts on travelers’ residential location choice. Residential location affects mode choice, and mode choice influences travel perception. Travel perception in the long term exerts an influence on life satisfaction. This example includes the time process and interactive process of different factors. Taniguchi et al. [45] established a process model among the environment factors (distance between home and work place, distance from home to bus station, etc.), psychological factors (travel satisfaction index), travel behavior (weekly use-frequency of each mode), goal achievement (if achieve or exceed the travel change goals), and future travel goals (travel frequency per week) to analyze interactions between different factors and influencing factors on voluntary change in travel behavior. Taniguchi and Fujii [46] tested an integrated process model of travel behavior change based on the theory of planned behavior, theories of habit, etc. Panel data collected before and after travel feedback program were used.

## 3. Methodology

### 3.1. Experiment Design of Information Intervention

In this study, the information intervention experiment was designed to guide respondents to reduce car use voluntarily through disseminating information about “health and environment benefits brought by non-motor vehicle traveling”, “air pollution caused by motor vehicle traveling”, and “improvement of green mode traveling”. Detailed intervention information was as follows:(1)According to World Health Organization report, walking more than 30 min every day make relative disease risk (including lung cancer, cardiovascular disease, cervical spondylosis, etc.) reduce by 22%. Cycling more than 30 min reduces relative disease risk by 28% [13].(2)According to statistics from the International Energy Agency, approximately 23% of global energy-related carbon dioxide emissions come from transport [47].(3)A bus is about 50 times capacity of a car. Fifty cars occupy 24 times the road area, consume 10 times the fuel, and exhaust 17 times the carbon dioxide of a bus vehicle. Car use increase will aggravate traffic congestion and cause more air pollution and carbon emissions.(4)More and more cities are suffering serious smog and haze. According to a report from the Chinese Academy of Sciences, four organic components in haze come from organic particles in the motor vehicle exhaust. In big cities, the main source of PM 2.5 is from the vehicle exhaust. Nitrogen Oxides and lead compounds in the exhaust are harmful to human central nervous system, resulting in sensory dysfunctions, hypertension, coronary heart disease, and even danger to life, especially for aged people and children [13].(5)Thirty-seven cities, including Beijing, Shanghai, Guangzhou, Wuhan, etc., have been actively creating “transit-oriented cities” and are committed to providing better public transport services. Traveling speed, waiting time, and congestion in vehicle are getting improved.(6)Bicycle lanes in Beijing are in continuous planning and construction. Riding environment is also improving. A bicycle-exclusive road between the Huilongguan and Shangdi region was built to attract more residents to use green mode.

The information intervention survey included four stages, T0, T1, I, and T2 (T2′), as shown in Figure 2.

In stage T0, individual characteristics, including age, gender, education, job, income, car ownership, public transport/bicycle availability, public transport pass availability, and attitude towards different transport modes were collected. In order to reduce participants’ burden and unwillingness to fill in questionnaire multiple times in a short period of time, T1 started one-week after T0.

In stage T1, participants were required to recall their recent trip. Recent trip attributes, including purpose, trip day, main mode during traveling (if one participant used more than two modes, main mode was the one used for the longest distance of the journey), travel duration, fee, travel time flexibility, travel companion, activities during traveling, and travel perception, were collected.

In stage I, participants in the test group received one message a day and continuous six days were needed to accept intervention information. Participants in the control group did not get any intervention information.

In stage T2 and T2′, participants in the control group and test group filled in their recent trip record including trip attributes and travel perception separately.

The data were derived from a web-based survey conducted from September to November 2019 in Beijing, the capital of China. Beijing had a population of 21.54 million by the end of 2019 and covers an area of 16, 410 km^2^. Available travel modes mainly include: conventional bus, bus rapid transit, customized bus, subway, private automobile, car sharing, taxi, online ride-hailing, private bike, and bike sharing. For a more detailed introduction to shared mobility (such as carsharing, bike-sharing, and carpooling), customized bus, and online ride-hailing, please refer to Adam and Susan [11], Liu and Ceder [48], and Chen et al. [49]. The mode share of public transport (including conventional bus, bus rapid transit, customized bus, and subway) was approximately 50% in 2017. Mode share of automobile (including private automobile, car sharing, taxi, and online ride-hailing) and bike (including private bike and shared bike) were about 30% and 10%, respectively. Beijing implemented license plate lottery since 2011 and residents had to win a license lottery to buy a new car. In order to fight against traffic congestion and air pollution, car use restriction policy had been implemented since 2008 and new energy vehicles purchase were encouraged in recent years.

The survey was conducted by a professional questionnaire company. Questionnaires were randomly issued to local residents in Beijing and permanent residents who live or work in Beijing for more than 6 months but without local household registration. According to the survey process depicted in Figure 2, 2335 valid samples were obtained in stage T0. Each additional round of survey would result in a certain percentage of sample loss. T1 was distributed among 2335 samples of T0, and finally 2118 valid samples were collected. Of the 2118 valid samples in stage T1, 318 participants were randomly chosen as the control group to collect T2, and 213 valid samples were finally obtained. The remaining 1800 participants in T1 were put into the test group used for receiving intervention information, and 1086 valid samples were finally collected in stage T2′.

### 3.2. Experiment Design of Public Transport Service Improvement

The whole process of traveling by public transport includes walking/cycling/driving to access station, waiting at station, staying in vehicle, transferring to other lines, staying in vehicle, and walking/cycling/driving to destination from egress station, as shown in Figure 3. The travel efficiency of taking public transport involved access time, waiting time, in-vehicle time, the number of transfers, and egress time. The travel comfort was mainly related with in-vehicle comfort.

In this study, the improvement of public transport service was reflected from five factors. Accessibility was increased to reduce the access/egress time. Next-bus service was provided to shorten the waiting time at public transport stations. Exclusive bus lane was set to decrease the in-vehicle time. Public transport network was optimized to reduce the number of transfers. Operation plan was adjusted according to the travel demand to increase in-vehicle comfort. According to the above five factors, different travel scenarios were designed. Analysis of commute travel characteristics in Beijing and typical regions [50] released by the Beijing Transport Institute showed that the average commute distance was 12.4 km within 6th ring of Beijing. So, the travel distance in the scenarios was set as 13 km. Access and egress time was set as 5, 10, and 15 min according to the 300 m and 500 m coverage of public transport service. Waiting time was set as 2, 6, and 10 min. Based on the Gaode map (amap.com), traveling 13 km by public transport took about 35–70 min at different time periods. In-vehicle time was set as 30, 45, and 60 min to indicate that public transport service improved. The number of transfers was set as 0 and 1. In-vehicle comfort was set as two levels, comfortable and crowded. Table 3 showed the detailed public transport service indicators and corresponding service levels.

Mixed-level uniform design was used to design travel scenarios. Table 4 presented the results of mixed-level uniform design. Six scenarios were different in service level and improvement aspects of public transport service.

Public transport cost was set as a fixed range, 1–4 yuan. Besides the public transport, the choice set of travel mode also included driving a car and taking a taxi. Time and costs of driving a car and taking a taxi was set as fixed values according to actual situation, as shown in Table 5. This experiment included two stages, R1 and R2. In stage R1, participants’ trip attributes and travel perception in recent trip were collected. In stage R2, trip attributes and travel perception in the six different travel scenarios were obtained.

The data were also derived from a web-based survey conducted in November 2019 in Beijing. The questionnaire R1 and R2 was released in 2335 samples of T0 in one time and 1013 valid samples were finally obtained.

### 3.3. Process Model

In this study, the process model was introduced to analyze mutual relations among transport interventions, mode choice, and travel perception in the time process. Figure 4a,b showed the process model framework of information intervention and public transport service improvement. The detailed variable description and model interpretation were presented in the next part, Section 4.

## 4. Result Analysis

### 4.1. Process Model of Information Intervention

The data of participants’ mode choice and travel perception before (T1) and after (T2 and T2′) intervention were collected. Individual characteristics were collected in stage T0. On the basis of these data, a process model among information intervention, mode choice, and travel perception would be proposed to explore their mutual relations.

#### 4.1.1. Variables and Descriptions

Mode choice in stage T1 affected participants’ travel perception. Travel perception in stage T1 had impacts on subsequent mode choice, namely mode choice in T2 (control group) and T2′ (test group). Information intervention (binary variable) as an independent variable was added into the mode choice model in stage T2 and T2′. Travel perception in stage T2 and T2′ was affected by mode choice and information intervention. Similarly, mode choice and travel perception in T2 and T2′ had impacts on travel behavior in subsequent stage T3, T4, …, Tn. Traveler’s mode choice and travel perception would become stable after a period of time, and finally reached a relatively stable state. In this study, research scope was limited to a short period after information intervention (within 1–3 weeks), namely within stage T1, T2 and T2′. Table 6 listed the detailed variables and descriptions of the information intervention process model. Individual characteristics, including gender, age, education, and income were included in the process model as control variables.

#### 4.1.2. Model Results

The process model of information intervention in stage T1, T2, and T2′ included three specific models, as shown in Table 6.

Model 1 was a multiple linear regression model that described the influence of mode choice on travel perception before intervention (stage T1). There were two dependent variables in Model 1, average score of cognitive evaluation (CE1) and average score of affective evaluation (AE1). Both were continuous variable. The independent variables were mode choice in stage T1 and individual characteristics. Mode choice in stage T1 included four specific variables, mod1_PT (public transport), mod1_bike (bike), mod1_walk (walk), and mod1_car (car). To avoid redundancy, car was selected as the reference mode.

Model 2 was a multinomial logit model. It represented the impact of travel perception in stage T1 and information intervention on mode choice in stage T2 and T2′. The dependent variable was mode choice in T2 and T2′ (nominal variable). Independent variables included travel perception in stage T1, whether intervention information was received, and individual characteristics. Because travel perception only was difficult to explain its effects on subsequent mode choices, mode choice and travel perception in stage T1 were combined to get travel perception for different travel modes, that is, travel perception by public transport, travel perception by bike, travel perception by walk, and travel perception by car. In addition, these four travel perceptions for different modes were divided into two intervals [−3, 0.5) and [0.5, 3] according to the average score of cognitive and affective evaluation. Therefore, travel perception in stage T1 was subdivided into four variables, PT_P, bike_P, walk_P, and car_P, which represented average score of travel perception by PT, bike, walk, and car was greater than 0.5 or not. Information intervention was a binary variable.

Similar to Model 1, Model 3 was also a multiple linear regression model. It was used to analyze the influence of information intervention and mode choice on travel perception in stage T2 and T2′. Average score of cognitive evaluation (CE2) and average score of affective evaluation (AE2) were dependent variables. The independent variables were mode choice, whether intervention information was received in stage T2 and T2′, and individual characteristics. Table 7 presented the regression results of the information intervention process model.

According to Table 7, in Model 1, compared with traveling by car, walking had more positive effects on cognitive and affective evaluation of travel perception. Public transport trips had negative effects on affective evaluation of travel perception. Model 1 indicated that travel perception during walking trips was more positive and it was easy to bring about negative emotions such as worry and anxiety during public transport trips. Age and education had significant impacts on travel perception. Aged people gave higher marks than the young and people with high levels of education tended to give low scores of travel perception.

According to the results in Model 2, travel perception in stage T1 had significant effects on mode choice in stage T2 and T2′. The more positive travel perception when using public transport in stage T1, the higher the probability of using public transport and walking in stage T2 and T2′, which were 2.8 and 4.3 times, respectively, than traveling by car. The more positive travel perception for bike trips in stage T1, the higher the probability of riding and walking in stage T2 and T2′. Similarly, if participants had more positive cognitive and affective evaluation during walking, they were more likely to travel by walking in stage T2 and T2′. If travelers felt more positive during car trips, they were less likely to use public transport and bike in stage T2 and T2′. Information intervention had positive impacts on public transport use and walking. In other words, the probabilities of travelers using public transport and walking increased after information intervention. However, the influence coefficients were quite small, which, indicating that information intervention had a very limited impact on public transport use and walking. It was found that public transport, bike, and walking were closely linked. Travelers who had positive evaluation of public transport and bike trips also tended to travel by walk, but travelers who had a positive evaluation of bike and walking trips did not increase the probability of travelling by public transport. Based on the above influence relationship, it was even more important to promote public transport use and improve the service level of public transport to enhance travelers’ positive perception of public transport trips. Income level had negative effects on bike and walking. The results in Model 2 could help to predicate travelers’ mode choice after implementing information intervention and evaluate information intervention according to transport mode shift from automobile to green modes.

In Model 3, compared with car trips, the influence of walking on travel perception was still positive. Public transport use had negative impacts on travel perception. Information intervention had direct influences on mode choice but no significant effects on travel perception. Age and education still had significant impacts on travel perception

The process model results of information intervention demonstrated that information intervention increased the probability of traveling by public transport and walking, but the impact was weak. Walking could increase positive aspects in travel perception. However, public transport trips had negative effects on travel perception. For travelers who had a high evaluation of travel perception during car trips, the probability of green mode (including public transport, bike, walking) use would decrease. Travelers who gave high marks to travel perception during public transport traveling, bike trips and walking would have a higher probability to keep the transport mode use or traveling by other green modes. The relationship among information intervention, mode choice, and travel perception is illustrated in Figure 5. Information intervention had direct impacts on mode choice, and mode choice affected travel perception. However, information intervention had no direct effect on travel perception.

### 4.2. Process Model of Public Transport Service Improvement

The data of participants’ mode choice and travel perception in recent trip (R1) and in different scenarios of public transport service improvement (R2) were collected. Individual characteristics were collected in stage T0. On the basis of these data, the process model among public transport service improvement, mode choice, and travel perception was proposed to explore their mutual relations.

#### 4.2.1. Variables and Descriptions

Mode choice in stage R1 affected participants’ travel perception. Travel perception in stage R1 had impacts on subsequent mode choice, namely mode choice in stage R2. In addition, mode choice in stage R2 was also affected by service level of public transport. Travel perception in R2 was influenced by mode choice and service level of public transport. The detailed variables and descriptions of the process model under public transport service improvement were shown in Table 8. Individual characteristics, including gender, age, education, and income, were included in the process model as control variables.

#### 4.2.2. Model Results

The process model of public transport service improvement in stages R1 and R2 included three specific models.

Model 1 was a multiple linear regression model which described the influence of mode choice on travel perception in stage R1. There were two dependent variables in Model 1, the average score of cognitive evaluation (CE1) and the average score of affective evaluation (AE1). Both were continuous variables. The independent variable was mode choice in stage R1 and individual characteristics. Mode choice in stage R1 included four specific variables, mod1_PT (public transport), mod1_bike (bike), mod1_walk (walk), and mod1_car (car). Car was selected as the reference mode.

Model 2 was a binary logit model, which represented the impact of travel perception in stage R1 and service level of public transport on mode choice in stage R2. Dependent variable was mode choice in stage R2 (nominal variable). Independent variables included travel perception in stage R1, service level of public transport in stage R2, and individual characteristics. Similar to Section 4.1.2, mode choice and travel perception in stage R1 were combined to get travel perception by public transport (PT_P), travel perception by bike (bike_P), travel perception by walk (walk_P), and travel perception by car (car_P). These four travel perceptions represented whether average score of travel perception by public transport, bike, walk, and car was greater than 0.5 or not. The service level of public transport in stage R2 included six scenarios, which were denoted by S1, S2, S3, S4, S5, and S6, respectively. The detailed service level was listed in Table 4.

Model 3 was also a multiple linear regression model used to analyze the impacts of public transport service level and mode choice on travel perception in stage R2. Average score of cognitive evaluation (CE2) and average score of affective evaluation (AE2) were dependent variables. Independent variables were the service level of public transport and mode choice in stage R2, and individual characteristics. Table 9 presented process model results of public transport service improvement.

According to Table 9, in Model 1, compared with traveling by car, public transport trips had negative effects on the affective evaluation of travel perception. Walking had more positive effects on cognitive and the affective evaluation of travel perception. The effect of cycling on travel perception was not significant. Similar to the results in the process model of information intervention, age and education had significant impacts on travel perception.

According to the results in Model 2, travel perception in stage R1 had significant effects on mode choice in stage R2. The more positive evaluation of travel perception when using public transport and bike in stage R1, the higher the probability of using public transport in subsequent stage R2. However, if travelers felt more positive during car trips in stage R1, they were less likely to use public transport in stage R2. Compared with scenario S4, scenario S1, S2, S3, S5, S6 all had significant positive effects on public transport use. The improvement aspects of public transport service level of S1, S2, S3, S5 and S6 were reflected by in-vehicle time reduction or the degree of comfort increase compared with S4, as shown in Table 10. The regression coefficient in S3 was the largest, indicating that travelers were most likely to choose public transport in S3. By comparing the differences in service levels of different scenarios and the regression coefficients in Model 2, it could be inferred that travelers gave more attention to in-vehicle time and degree of comfort when traveling by public transport. The probability of choosing public transport in S1, S2, S3, S5, and S6 was 4.0, 3.8, 5.2, 1.3, and 3.6 times that of choosing a car, respectively, compared with scenario S4. Gender, age, and income level had significant effects on mode choice. The female, aged, and low-income people more tended to choose public transport compared with the male, young, and high-income people.

In Model 3, the regression coefficients suggested that scenario S2, S3, and S6 had positive impacts on travel perception. In scenario S2, S3, S6, there were no crowding in vehicle and every passenger had a seat. It could be concluded that increased comfort in vehicle improve passengers’ travel perception significantly. Compared with traveling by car, public transport trips had negative effects on the affective evaluation of travel perception. It meant that car travelers felt more pleasant and enjoy traveling compared with traveling by public transport. Age and education still had significant impacts on travel perception

The process model results of public transport service improvement demonstrated that the improvement of public transport service significantly increased the probability of public transport use. Compared with car trips, walking could increase positive aspects in travel perception; however, public transport trips had negative effects on travel perception. For travelers who had a high evaluation of travel perception during car trips, the probability of public transport use would decrease. Travelers who gave high marks to travel perception during public transport traveling and bicycle trips would have a higher probability to travel by public transport. The relationship between public transport service improvement, mode choice, and travel perception is illustrated in Figure 6. In-vehicle time and comfort had significant impacts on public transport use. In-vehicle comfort had positive effects on travel perception.

## 5. Conclusions

This study focused on the influence of different transport intervention strategies, including information intervention and public transport service improvement on mode choice and travel perception. Based on the collected mode choice and travel perception data under these two intervention strategies, process models of information intervention and public transport service improvement were proposed to explore the relationship among transport interventions, mode choice, and travel perception.

The results from the process model of information intervention demonstrate that information intervention increased the probability of traveling by public transport and walking, but the impact was weak (the influencing coefficients were 0.015 and 0.007, respectively). Mode choice had significant effects on travel perception. Compared with car trips, walking could increase positive travel perception; however, public transport trips had negative effects on travel perception. The travel perception for different transport modes affected subsequent mode choice. For travelers who had a high evaluation of travel perception during car trips, the probability of green mode (including public transport, bike, walking) use would decrease. Travelers who gave high marks to travel perception during public transport traveling, bike trips and walking would have a higher probability to keep the transport mode use or traveling by other green modes. Information intervention had no significant effect on travel perception.

The results from the process model of public transport service improvement indicate that the improvement of public transport service significantly increased the probability of public transport use. The improvement in in-vehicle time and comfort could particularly enhance public transport mode choice dramatically. In-vehicle comfort improvement had positive effects on travel perception. The relationship between mode choice and travel perception were similar to that in the information intervention process model.

Policy suggestions were presented here in the light of the findings. (1) Information intervention had minor impacts on mode choice and no significant effect on travel perception. It was not suggested to only implement information intervention to change travelers’ mode choice. (2) Compared with traveling by car, walking had more positive impacts on travel perception. Promoting walking through traveler-oriented transport planning helped to enhance overall travel perception. (3) The improvement of public transport service, especially in-vehicle time reduction and in-vehicle comfort increase, helped to facilitate public transport use and increase travel perception.

Future studies may focus on the following: (1) Different traveler groups may give different responses to transport interventions strategies. Travelers can be classified into different groups according to their travel characteristics and individual attributes. Then, the effects of transport interventions strategies to different traveler groups can be analyzed. (2) Different intervention strategies are often implemented simultaneously. The combination of multiple transport interventions strategies should be further considered to analyze the comprehensive impacts on mode choice and travel perception. (3) In this study, only the short-term impacts of transport interventions on mode choice and travel perception were explored. In further studies, long-term impacts should be conducted and compared to short-term impacts.

## Figures and Tables

**Figure 1 ijerph-17-04258-f001:**
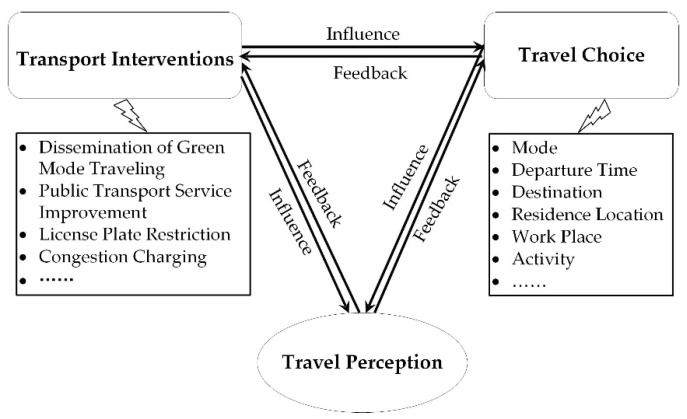
Relationship among transport interventions, travel choice, and travel perception.

**Figure 2 ijerph-17-04258-f002:**
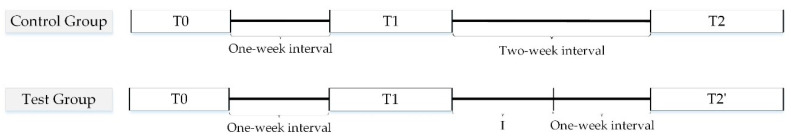
Arrangement of information intervention survey.

**Figure 3 ijerph-17-04258-f003:**
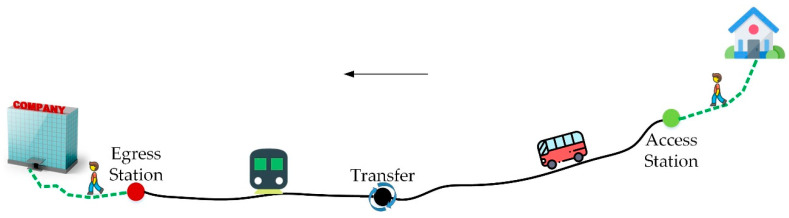
Sketch diagram of traveling by public transport.

**Figure 4 ijerph-17-04258-f004:**
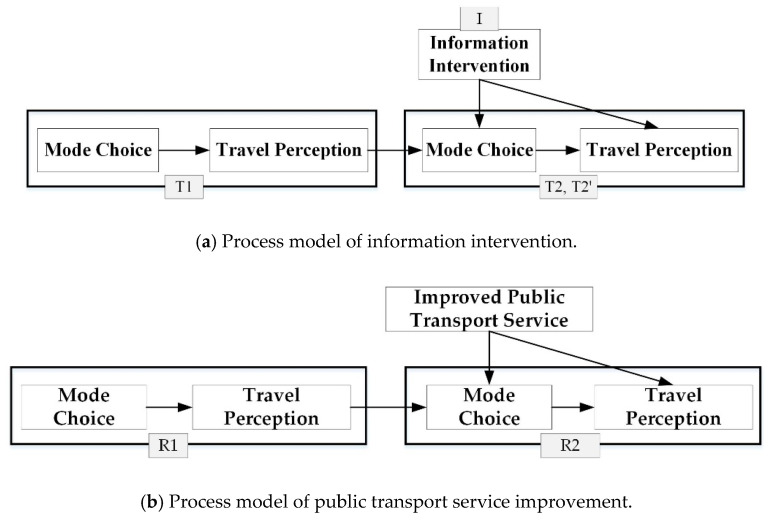
Framework of the process model.

**Figure 5 ijerph-17-04258-f005:**
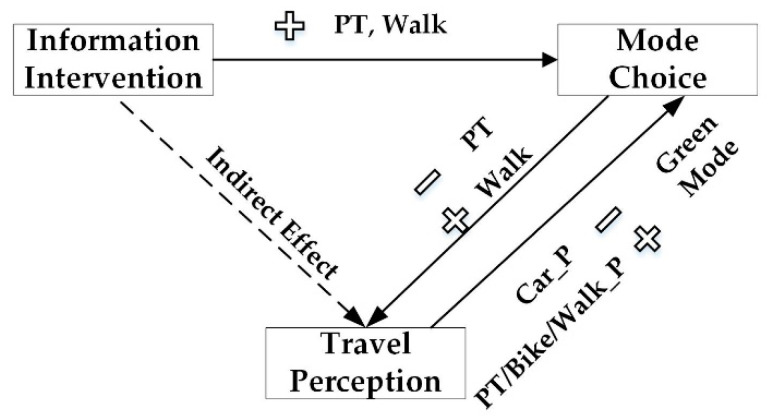
Relationship among information intervention, mode choice, and travel perception. (Note: 
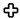
—increase; 
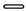
—decrease; PT—public transport; Car_P—perception of car trips; PT/Bike/Walk_P—perception of public transport/bike/walk trips.).

**Figure 6 ijerph-17-04258-f006:**
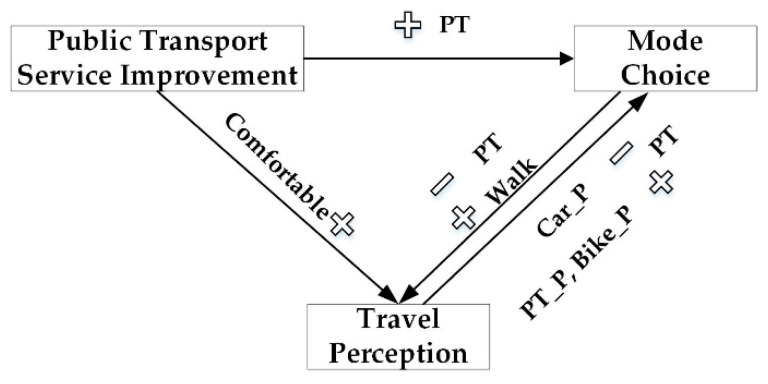
Relationship among public transport service improvement, mode choice, and travel perception. (Note: 
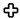
—increase; 
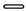
—decrease; PT—public transport; Car_P—perception of car trips; PT _P, Bike _P—perception of public transport and bike trips.).

**Table 1 ijerph-17-04258-t001:** Main strategies of transport interventions.

Transport Interventions	Intervention Type	Examples
Physical Change	Hard Measures	High occupancy vehicle and toll lanes
	Soft Measures	Public transport service improvement Shared mobility service providing (such as minibus for first-and last-mile connections, late night bus, and paratransit) Walking and riding environment improvement
Legal Policies	Hard Measures	License-plate lottery License-plate restriction
Economic Policies	Hard Measures	Congestion charging Taxation of cars and fuel
	Soft Measures	Discounted transfer Fare-free public transport service
Information and Education	Soft Measures	Public information campaigns Giving feedback about consequences of transport projects

**Table 2 ijerph-17-04258-t002:** Satisfaction with Travel Scale (STS).

**Cognitive Evaluation**
Travel was worst (−3)—best I can think of (3)
Travel was low (−3)—high standard (3)
Travel worked poorly (−3)—worked well (3)
**Affective Evaluation**
Tired (−3)—Alert (3)
Bored (−3)—Enthusiastic (3)
Fed up (−3)—Engaged (3)
Time pressed (−3)—Relaxed (3)
Worried I would not be in time (−3)—Confident I would be in time (3)
Stressed (−3)—Calm (3)

**Table 3 ijerph-17-04258-t003:** Public transport service indicators and corresponding service levels.

Access-Egress Time (Minute)	Waiting Time (Minute)	In-Vehicle Time (Minute)	Number of Transfers	Degree of Comfort
5	2	30	0	Comfortable (every passenger has a seat)
10	6	45	1	Crowded
15	10	60	—	—

**Table 4 ijerph-17-04258-t004:** Results of mixed-level uniform design.

Scenario	Service Level of Public Transport	Improvement Aspects of Public Transport Service
S1	Access-egress time: 5 min; Waiting time: 2 min; In-vehicle time: 45 min; One transfer; Crowded	Accessibility + next-bus service + exclusive bus lane
S2	Access-egress time: 5 min; Waiting time: 6 min; In-vehicle time: 60 min; No transfer; Comfortable	accessibility + next-bus service + network + operation plan
S3	Access-egress time: 10 min; Waiting time: 10 min; In-vehicle time: 30 min; One transfer; Comfortable	accessibility + exclusive bus lane + operation plan
S4	Access-egress time: 10 min; Waiting time: 2 min; In-vehicle time: 60 min; No transfer; Crowded	accessibility + next-bus service + network
S5	Access-egress time: 15 min; Waiting time: 6 min; In-vehicle time: 30 min; One transfer; Crowded	next-bus service + exclusive bus lane
S6	Access-egress time: 15 min; Waiting time: 10 min; In-vehicle time: 45 min; No transfer; Comfortable	exclusive bus lane + network + operation plan

**Table 5 ijerph-17-04258-t005:** Schematic of scenario 6.

Indicator	Public Transport	Car	Taxi
Time	15 min to access to and egress from station 10 min to wait 45 min to stay in vehicle	30 min to drive 5 min to park	7 min to wait 30 min to stay in taxi
Transfer	No transfer		
Fee	1–4 yuan	Fuel fee: 7 yuan Parking fee: 15 yuan	35 yuan
Degree of Comfort of Public Transport	Comfortable 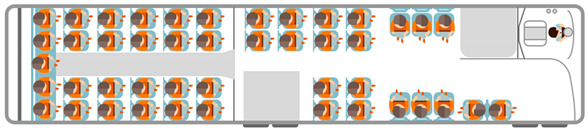		

**Table 6 ijerph-17-04258-t006:** Variable descriptions of information intervention process model.

Model	Variable	Specific Variable	Description	Encoded Value
Model 1	Mode Choice (T1)	mod1_PT	Choose PT or not	0—not choose 1—choose
		mod1_bike	Choose bike or not	0—not choose 1—choose
		mod1_walk	Choose walk or not	0—not choose 1—choose
		mod1_car (Reference)	Choose car or not	0—not choose 1—choose
	Individual Characteristics	gender		0—male; 1—female
		age		Ordinal variable
		education		Ordinal variable
		income		Ordinal variable
	Travel Perception (T1)	CE1	Average score of cognitive evaluation	Continuous variable
		AE1	Average score of affective evaluation	Continuous variable
Model 2	Travel Perception (T1)	PT_P	Average score of travel perception by PT ≥0.5 or not	0—<0.5 1—≥0.5
		bike_P	Average score of travel perception by bike ≥0.5 or not	0—<0.5 1—≥0.5
		walk_P	Average score of travel perception by walk ≥0.5 or not	0—<0.5 1—≥0.5
		car_P	Average score of travel perception by car ≥0.5 or not	0—<0.5 1—≥0.5
	Information Intervention	if_inter	Get intervention information or not	0—No 1—Yes
	Individual Characteristics		Same as that in Model 1, omitted here.	
	Mode Choice (T2, T2′)	mod_2	Mode choice in T2 (T2′)	PT Bike Walk Car (Reference)
Model 3	Information Intervention	if_inter	Get intervention information or not	0—No 1—Yes
	Mode Choice (T2, T2′)	mod2_PT	Choose PT or not	0—not choose 1—choose
		mod2_bike	Choose bike or not	0—not choose 1—choose
		mod2_walk	Choose walk or not	0—not choose 1—choose
		mod2_car (Reference)	Choose car or not	0—not choose 1—choose
	Individual Characteristics		Same as that in Model 1, omitted here.	
	Travel Perception (T2, T2′)	CE2	Average score of cognitive evaluation	Continuous variable
		AE2	Average score of affective evaluation	Continuous variable

Note: PT was the abbreviation of public transport.

**Table 7 ijerph-17-04258-t007:** Regression results of information intervention process model (N = 1086).

Independent Variable		Dependent Variable						
		Model 1		Model 2			Model 3	
		Travel Perception (T1)		Mode Choice (T2, T2′)			Travel Perception (T2, T2′)	
		CE1	AE1	PT	Bike	Walk	CE2	AE2
Mode Choice (T1)	mod1_PT	−0.114	−0.110 **	—	—	—	—	—
	mod1_bike	0.195	0.327	—	—	—	—	—
	mod1_walk	0.406 **	0.497 **	—	—	—	—	—
Individual Characteristics	gender	0.051	0.083	−0.090	−0.229	0.014	0.110	−0.006
	age	0.094 **	0.146 **	−0.258	0.055	−0.324	0.029	0.095 **
	education	−0.118 *	−0.205 **	−0.307	−0.409	−0.386	−0.227 **	−0.206 **
	income	−0.064	0.001	−0.004	−0.527 *	−0.415 *	0.068	0.100
Travel Perception (T1)	PT_P	—	—	1.031 **	−0.123	1.450 **	—	—
	bike_P	—	—	0.724	2.566 **	1.777 **	—	—
	walk_P	—	—	−0.020	0.536	2.413 **	—	—
	car_P	—	—	−0.803 **	−1.036 *	−0.244	—	—
Information Intervention	if_inter	—	—	0.015 *	0.007 *	0.004 *	0.013 *	0.012
Mode Choice (T2, T2′)	mod2_PT	—	—	—	—	—	0.216	−0.099 **
	mod2_bike	—	—	—	—	—	0.328	0.466
	mod2_walk	—	—	—	—	—	0.461 **	0.670 **
Constant		1.854 **	1.456 **	1.470 **	0.612	0.359	1.787 **	1.251 **
R Square		0.23	0.26	0.42			0.22	0.25

Note: ** —significant at 0.05 level; * —significant at 0.1 level.

**Table 8 ijerph-17-04258-t008:** Variable descriptions of process model for public transport service improvement.

Model	Variable	Specific Variable	Description	Encoded Value
Model 1	Mode Choice (R1)	mod1_PT	Choose PT or not	0—not choose 1—choose
		mod1_bike	Choose bike or not	0—not choose 1—choose
		mod1_walk	Choose walk or not	0—not choose 1—choose
		mod1_car (Reference)	Choose car or not	0—not choose 1—choose
	Individual Characteristics	gender		0—male; 1—female
		age		Ordinal variable
		education		Ordinal variable
		income		Ordinal variable
	Travel Perception (R1)	CE1	Average score of cognitive evaluation	Continuous variable
		AE1	Average score of affective evaluation	Continuous variable
Model 2	Travel Perception (R1)	PT_P	Average score of travel perception by PT ≥0.5 or not	0—<0.5 1—≥0.5
		bike_P	Average score of travel perception by bike ≥0.5 or not	0—<0.5 1—≥0.5
		walk_P	Average score of travel perception by walk ≥0.5 or not	0—<0.5 1—≥0.5
		car_P	Average score of travel perception by car ≥0.5 or not	0—<0.5 1—≥0.5
	Service Level of PT (R2)	S1	accessibility + next-bus service + exclusive bus lane	0—not scenario 1 1—scenario 1
		S2	accessibility + next-bus service + network + operation plan	0—not scenario 2 1—scenario 2
		S3	accessibility + exclusive bus lane + operation plan	0—not scenario 3 1—scenario 3
		S4 (Reference)	accessibility + next-bus service + network	0—not scenario 4 1—scenario 4
		S5	next-bus service + exclusive bus lane	0—not scenario 5 1—scenario 5
		S6	exclusive bus lane + network + operation plan	0—not scenario 6 1—scenario 6
	Individual Characteristics		Same as that in Model 1, omitted here.	
	Mode Choice (R2)	mod_2	Mode choice in stage R2	PT car(Reference)
Model 3	Service Level of PT (R2)	Same as service level of public transport in Model 2, omitted here.		
	Mode Choice (R2)	mod2_PT	Choose PT or not	0—not choose 1—choose
		mod2_car (Reference)	Choose car or not	0—not choose 1—choose
	Individual Characteristics		Same as that in Model 1, omitted here.	
	Travel Perception (R2)	CE2	Average score of cognitive evaluation	Continuous variable
		AE2	Average score of affective evaluation	Continuous variable

Note: PT was the abbreviation for public transport.

**Table 9 ijerph-17-04258-t009:** Process model results of public transport service improvement (N = 1013).

Independent Variable		Dependent Variable				
		Model 1		Model 2	Model 3	
		Travel Perception (R1)		Mode Choice (R2)	Travel Perception (R2)	
		CE1	AE1	mod2_PT	CE2	AE2
Mode Choice (R1)	mod1_PT	0.060	−0.119 **	—	—	—
	mod1_bike	0.085	0.236	—	—	—
	mod1_walk	0.556 **	0.693 **	—	—	—
Individual Characteristics	gender	0.096	0.085	0.179 **	0.137 *	0.083
	age	0.008 **	0.095 **	0.236 **	0.105 **	0.168 **
	education	−0.099 **	−0.195 **	0.758	−0.177 **	−0.175 **
	income	0.046	0.109	−0.101 **	0.062	0.069
Travel Perception (R1)	PT_P	—	—	0.345 **	—	—
	bike_P	—	—	0.334 **	—	—
	walk_P	—	—	0.457	—	—
	car_P	—	—	−0.748 **	—	—
Service Level of PT (R2)	S1	—	—	1.408 **	0.010	−0.044
	S2	—	—	1.371 **	0.315 **	0.239 **
	S3	—	—	1.670 **	0.367 **	0.347 **
	S5	—	—	0.261 **	-0.046	0.021
	S6	—	—	1.308 **	0.246 **	0.326 **
Mode Choice (R2)	mod2_PT	—	—	—	−0.263	−0.352 **
Constant		1.461 **	0.939 **	−0.876 **	1.390 **	1.097 **
R Square		0.22	0.28	0.35	0.26	0.31

Note: ** —significant at 0.05 level; * —significant at 0.1 level.

**Table 10 ijerph-17-04258-t010:** Service level comparison of different scenarios.

Scenario	Access-Egress Time (Minute)	Waiting Time (Minute)	In-Vehicle Time (Minute)	Number of Transfers	Degree of Comfort	Total Time (Minute)	Percentage of Choosing PT
S1	5	2	45	1	Crowded	52	69.4%
S2	5	6	60	0	Comfortable	71	68.6%
S3	10	10	30	1	Comfortable	50	74.5%
S4	10	2	60	0	Crowded	72	37.6%
S5	15	6	30	1	Crowded	51	43.7%
S6	15	10	45	0	Comfortable	70	67.5%

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
