# Peer review of "Exploring the Relationship between Transport Interventions, Mode Choice, and Travel Perception: An Empirical Study in Beijing, China"

_ijerph, 2020, doi:10.3390/ijerph17124258_

Round 1

Reviewer 1 Report

Literature Review

The literature review of this article is approximately comprehensive. It is suggested to increase the review of research methods.

Methodology

(1)About the part of information interaction to explore the relationship between Mode Choice and Transport Perception.

In the transport perception questionnaire survey, there is a palpable gap between the number of samples selected in T2 stage and T2 'stage, which may have a considerable impact on the accuracy of the research results.

In terms of the multinomial logit model established in Model 2, the explain of the methodology application is not specific.

(2)About the method of exploring the influence of mode choice on travel perception in all the models established in this paper.

It is suggested to explore the relationship between mode choice and travel perception after controlling individual characteristics variables such as gender / income / education.

Anyway, this study is interesting.

Author Response

Thanks for your suggestions which have made great contributions to the revision of this paper. Based on your comments and suggestions, we have made detailed modifications.

Please see the response letter in the attachment.

Reviewer 2 Report

This study examines the influencing factors of transport interventions and traveler choices on traveler perceptions. 

Table 1 - The "soft measures" for public transport improvement that are discussed in this paper are rather limited (e.g., free public transit and information campaigns). What about the role of shared mobility filling gaps in public transit service (e.g., first- and last- mile connections, late night service, paratransit etc.). For more information on shared modes:

https://escholarship.org/uc/item/9678b4xs

https://www.planning.org/publications/report/9107556/

Even if this is not a core component of the study, some discussion of shared modes and what modes are available in the context of the local environment where the survey was conducted should be included.  

"such as mode share, new car sales, energy saving, and ridesharing system adoption"

The authors are encouraged not to use the term "ridesharing". For more information, please refer to SAE International J3163. 

https://www.sae.org/standards/content/j3163_201809/'

Methods - How was outreach for the online survey conducted? 

Author Response

we would like to thanks for your suggestions which have made great contributions to the revision of this paper. Based on your comments and suggestions, we have made detailed modifications.

Please see the response letter in the attachment.

Round 2

Reviewer 2 Report

Authors have addressed reviewer comments. One question ...

"Available travel modes mainly include: conventional bus, bus rapid transit,
customized bus, subway, private automobile, car sharing, taxi, online ride-hailing, private bike, and shared bike." For readers who are not familiar with these services/terms, should there be a reference to other work with information on shared mobility/definitions? For example (Chapter 1): https://www.planning.org/publications/report/9107556/

Author Response

The authors would like to thank again for your suggestions. We have made detailed modifications  according to your comments as follows.

Please see the response letter in the attachment.
